# Electrical and Humidity-Sensing Properties of Impedance-Type Humidity Sensors that Were Made of Ag Microwires/PPy/SnO$_2$ Ternary Composites

**Pi-Guey Su \*** and **Ping-Hsiang Lu**

Department of Chemistry, Chinese Culture University, Taipei 111, Taiwan; scott21307@gmail.com
\* Correspondence: spg@faculty.pccu.edu.tw; Tel.: +886-2-2861-0511 (ext. 25332)

**Abstract:** Impedance-type humidity sensors were fabricated via one-step UV-irradiation photopolymerization of Ag microwires (Ag MWs), polypyrrole (PPy) and SnO$_2$ ternary composite (Ag MWs/PPy/SnO$_2$) films on an alumina substrate. X-ray diffractometry (XRD), Fourier transform infrared spectroscopy (FTIR), and scanning electron microscopy (SEM) coupled with an energy dispersive X-ray (EDX) elemental mapping were used to analyze the morphology, structure, and composition of Ag MWs/PPy/SnO$_2$ ternary composite films. Microstructural observations revealed that the Ag MWs were embedded, and PPy formed on the surface of the Ag MWs/PPy/SnO$_2$ ternary composite film. The effects of the addition amounts of loading of Ag and PPy on the electrical and humidity-sensing properties of the Ag MWs/PPy/SnO$_2$ ternary composite films were investigated. The impedance-type humidity sensor based on Ag MWs/PPy/SnO$_2$ ternary composite film containing 6 mg of Ag and 0.1 g of PPy had the highest sensitivity and an acceptable linearity over the RH ranged from 10% to 90% RH, a low hysteresis, a fast response time, and long-term stability. This technique is useful for practical application because its fast and ease of fabrication. The ions (H$_3$O$^+$) that dominate the impedance changed with relative humidity (RH) for the humidity sensor that based on Ag MWs/PPy/SnO$_2$ ternary composite film was analyzed using complex impedance spectra.

**Keywords:** humidity sensor; Ag microwires; polypyrrole (PPy); SnO$_2$; ternary composite

## 1. Introduction

Measuring and controlling humidity is an important issue because humidity is necessary in controlling the quality of productions in fabrication processes and maintaining human health in a comfortable environment [1–3]. Therefore, fabrication of humidity sensors mostly focused on high sensitivity over wider humidity level, low cost, fast response/recovery times, good reproducibility, ease of fabrication, and long-term stability of the sensors [4,5]. Various materials have been fabricated as humidity sensors including polymers [6–10], ceramics [11–14], and composite materials [5,15–19]. Among these materials, ceramics are favored materials in the humidity-sensing because they had the advantages such as good heat resistance, cost-effectiveness, water resistance at high humidity, and high stability in a wider humidity-working range [5]. Among various metal oxide sensors, most of the development work was centered on SnO$_2$ material due to its advantages such as lower working temperature, excellent electrochemical stability, high sensitivity to water vapor in air, non-toxicity, corrosion resistance, low cost and long lifetime [20,21], which makes it can be used as a candidate material for humidity detection. SnO$_2$, a n-type semiconducting material, was the most studied ceramic humidity-sensing material because it exhibited inherent chemical, physical stability, and helpful H$_2$O adsorption ability [5,22], but did not have obvious resistance changes at higher relative humidity (RH) due to its low conductivity [23], limiting its practical applications. The sensitivity of the humidity

sensors based on $SnO_2$-based materials depended upon the reactions between the surfaces of $SnO_2$ and water molecules. Thus, many researchers focused on tailoring the microstructure and morphology of $SnO_2$ structures [22,24]. Modifying the $SnO_2$ with various metal oxides, nanocarbon materials, noble metal particles, and alkali ions for enhancing the humidity-sensing properties of $SnO_2$ are also an important approach such as ZnO [25], $TiO_2$ [26], $WO_3$ [27], $WS_2$ [28], reduced graphene oxide [23], Ag [29], Sb [30], $K^+$ [31], and $Li^+$ [32]. Among them, Ag particles has recently attracted much attention because of their high stability, good conductivity, and biocompatibility as the dopant for enhancing the humidity-sensing properties [29,33].

Conducting polymers (CPs) have attracted much attention in fabrication of humidity sensors because that they exhibited unique advantages such as lots of surface area and good conductivity for enhancing the performance of sensor devices [34]. Polypyrrole (PPy) was the most studied conducting polymer material owing to its surface properties can easily be modified by various dopants during synthesis, and it is easily synthesized. Moreover, Ag-PPy nanocomposites in various microstructures have also been synthesized using a UV-irradiation technique using silver nitrate ($AgNO_3$) as an oxidizing agent (electron acceptor) [35–38]. Li et al. [35] fabricated Ag–polypyrrole coaxial nanocables. Lu et al. [36] and Zhao et al. [37] fabricated Ag nanoparticles decorated PPy. This technique had many advantages such as simple one-step room-temperature reaction, reducing the time of pyrrole polymerization and directly depositing metal nanoparticles and conducting polymers composite onto insulating substrates [36,39]. Kiani et al. [40] fabricated a $NH_3$ gas sensor based on PPy fiber-Ag nanocomposite using in-situ chemical synthesis method. In our previous study [41], Ag microparticels/PPy/$TiO_2$ composite was fabricated using UV-irradiation photopolymerization technique for sensing humidity. No attempt has been made to fabricate a humidity sensor based on Ag microwires (Ag MWs), PPy, and $SnO_2$ ternary composite (Ag MWs/PPy/$SnO_2$) using UV-irradiation photopolymerization. Moreover, one-dimensional (1D) nanostructured materials—such as wires, tubes, and rods—exhibited many advantages such as unique electrical and mechanical properties, high porosity, and lots of surface area so that they have been used in gas sensors [42,43]. Therefore, in this study, an impedance-type humidity sensor that based on Ag MWs/PPy/$SnO_2$ ternary composite thin film using a simple one-step room temperature UV-irradiation photopolymerization technique. The characterization of the Ag MWs/PPy/$SnO_2$ films were analyzed by scanning electron microscopy (SEM) coupled with an energy dispersive X-ray (EDX) elemental mapping, Fourier transform infrared spectroscopy (FTIR), and X-ray diffraction (XRD). The humidity sensing and electrical properties of the Ag MWs/PPy/$SnO_2$ ternary composite films in which various amounts of added Ag and PPy, and the time of the UV-irradiation photopolymerization were investigated. The ions dominating the resistance changes of the Ag MWs/PPy/$SnO_2$ ternary composite film varied with relative humidity (RH) was investigated using complex impedance spectra.

## 2. Experimental Methods

### 2.1. Materials

The chemicals were used as received without further purification: stannous chloride dihydrate ($SnCl_2·2H_2O$, 98%, Sigma-Aldrich, St. Louis, MO, US), hydrochloric acid (HCl, 37%, Sigma-Aldrich, St. Louis, MO, US), pyrrole (Py, 98%, Sigma-Aldrich, St. Louis, MO, US), $AgNO_3$ (99%, Mallinckrodt Baker Inc., Phillipsburg, NJ, US), and ethanol ($C_2H_5OH$, 95%, Sigma-Aldrich, St. Louis, MO, US).

### 2.2. Fabrication of Humidity Gas Sensors Based on Ag MWs/PPy/$SnO_2$ Ternary Composite and Measurement of Their Humidity-Sensing Properties

The precursor solution of the $SnO_2$ was prepared via added 0.225 mmol $SnCl_2·2H_2O$ and 150 μL HCl (36–38%) into 10 mL deionized water (DIW) and the as-prepared mixed solution was stirred at 400 rpm speed for 2 h. The PPy precursor solution was prepared via added $AgNO_3$ (Mallinckrodt Baker Inc., Phillipsburg, NJ, USA) to the Py in ethanol and sonicated the mixture to completely dissolve the

$AgNO_3$. Mixing the precursor solution of the $SnO_2$ and the PPy in various mixing compositions and sonicated the as-prepared mixed precursor solution for uniformly dispersing them. Then, as-prepared uniformly mixed precursor solution was drop-coated on an alumina substrate with five pairs of interdigitated Au electrodes. The films were polymerized using UV light (254 nm) for 20 min, resulting black films. The fabrication of pure $SnO_2$ was as the same our previous study [44]. Figure 1a plots the structure of an as-prepared impedance-type humidity sensor. Figure 1b shows the humidity-sensing measurement system for sensors testing. A divided humidity generator system was used to control the generation of required humidity conditions for testing. The total flow rate of the mixing dry and humid air was 10 L/min. A standard humidity hygrometer (with an accuracy of ±0.1% RH) was used to calibrate the required RH values of the measurement system. An LCZ meter was used to measure the impedance of the as-prepared humidity sensors as function of the RH in a temperature-controlled testing chamber. The experimental measurement conditions were at 1 kHz frequency, 1 V applied voltage, and 25 °C ambient temperature, various humidities ranged from 10% to 90% RH.

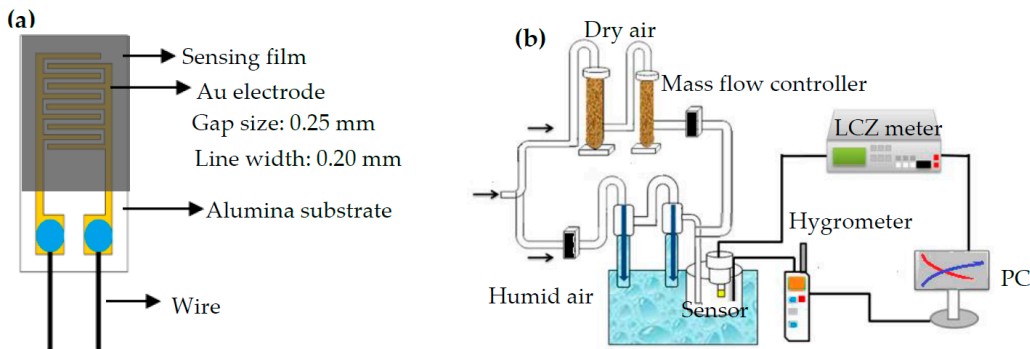

**Figure 1.** (**a**) Structure of humidity sensor and (**b**) the impedance measurement of humidity sensors and humidity atmosphere controller.

### 2.3. Characterization of Ag MWs/PPy/SnO₂ Ternary Composites

The composition and morphologies of the Ag MWs/PPy/SnO₂ ternary composite films coated on an alumina substrate was investigated using a X-ray diffraction (XRD) using Cu $K_\alpha$ radiation (Lab XRD-6000, Shimadzu Corporation, Kyoto, Japan), scanning electron microscope (SEM, JSM-7600F, JEOL, Tokyo, Japan) coupled with an energy dispersive X-ray (EDX) elemental mapping, and Fourier transform infrared spectroscopy (FTIR, Nicolet 380, Thermo Fisher, Scientific, Taipei, Taiwan).

## 3. Results and Discussion

### 3.1. Characteristics of Ag MWs/PPy/SnO₂ Ternary Composite Films

#### 3.1.1. XRD Characterization

Figure 2 plots the XRD spectra of the pristine $SnO_2$, standalone Ag/PPy and Ag MWs/PPy/SnO₂ ternary composites that were prepared by one-step UV-induced photopolymerization processes. The XRD patterns of pristine $SnO_2$ included main peaks at 2θ = 26°, 33°, 37°, 51°, 54°, 65°, 66°, 71°, and 78°, corresponding to the (110), (101), (200), (211), (220), (112), (301), (202), and (321) planes of tetragonal phase $SnO_2$, indicating the formation of $SnO_2$ crystals [45,46]. For the XRD patterns of Ag/PPy, a broad diffraction peak (104) appears at 24°, revealing the formation of amorphous PPy [47]. Another four sharp diffraction peaks at 2θ = 38°, 44°, 64°, and 77°, corresponded to the (111), (200), (220), and (311) planes of Ag, indicating the face centered cubic (FCC) structure of Ag [48]. XRD patterns of Ag MWs/PPy/SnO₂ ternary composite had the main characteristic peaks of pristine $SnO_2$ and standalone Ag/PPy, respectively, albeit with relatively low intensities, indicating the simultaneous existence of $SnO_2$, PPy, and Ag in the ternary composites.

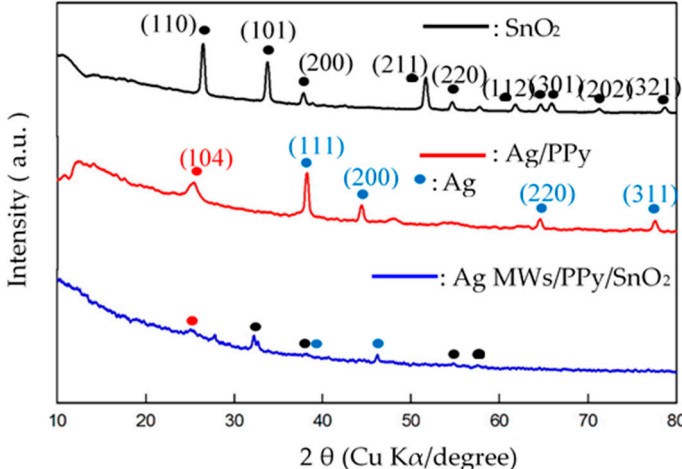

**Figure 2.** XRD patterns of pristine SnO$_2$, standalone Ag/PPy, and Ag MWs/PPy/SnO$_2$ ternary composite that was fabricated using one-step UV-irradiation photopolymerization technique.

### 3.1.2. IR Spectra

Figure 3 plots the IR spectra of the pristine SnO$_2$, standalone Ag/PPy and Ag MWs/PPy/SnO$_2$ ternary composites that were prepared by one-step photopolymerization processes. For IR spectrum of pristine SnO$_2$, the band at 620 cm$^{-1}$ was observed, corresponding to the anti-symmetric and symmetric vibrations of Sn–O–Sn [49]. For IR spectrum of standalone Ag/PPy, the characteristic N-H stretching, C=C stretching, C-N stretching, and N-H bands of PPy corresponding to at 3300, 1600, 1485, and 1030 cm$^{-1}$ were observed [50]. IR spectrum of Ag MWs/PPy/SnO$_2$ ternary composite exhibited the main characteristic absorption peaks of pristine SnO$_2$ and standalone Ag/PPy, respectively, providing direct evidence that PPy and SnO$_2$ were present in the Ag MWs/PPy/SnO$_2$ ternary composite.

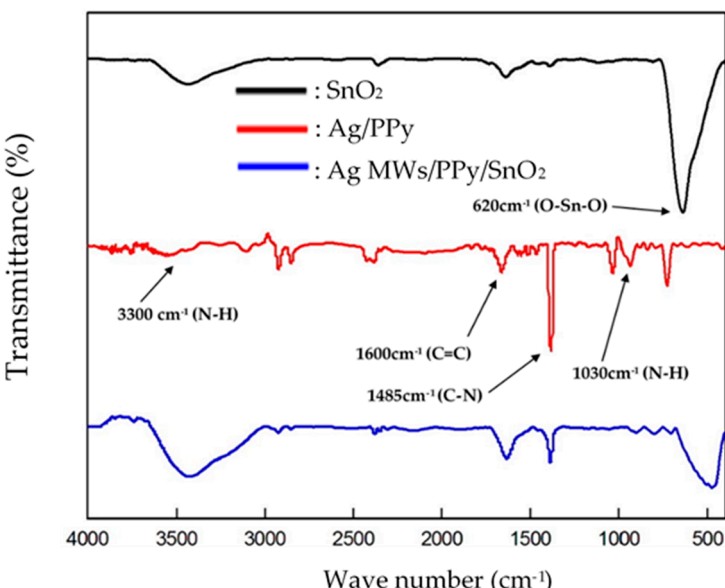

**Figure 3.** IR spectra of pristine SnO$_2$, standalone Ag/PPy and Ag MWs/PPy/SnO$_2$ ternary composite that was fabricated using one-step UV-irradiation photopolymerization technique.

### 3.1.3. SEM Analyses

Figure 4a shows the pristine SnO$_2$ film, the SnO$_2$ particles were micro-spherical agglomerates with sizes ranging from 0.1 to 1 μm. Moreover, these particles obviously aggregated to form various unfixed shapes of massive lamination films. Figure 4b shows Ag MWs/PPy/SnO$_2$ ternary composite

film in low-magnification image, the PPy glued the $SnO_2$ particles together and formed on the surface of Ag MWs/PPy/$SnO_2$ ternary composite film. Notably, lots of wires were observed from the fracture of image marked by white arrows. Figure 4c shows the high-magnification image of the fracture of image of Figure 4b, the wires with diameters ranging from 0.22 μm to 0.25 μm were embedded and wrapped and more dispersed inside the Ag MWs/PPy/$SnO_2$ ternary composite. In order to explore the compositions of the Ag MWs/PPy/$SnO_2$ ternary composite film, the EDX elemental mapping (Figure 4d–i) of the area in the image of the fracture in Figure 4b was obtained, showing the presence of C, N, O, Sn, and Ag components in the Ag MWs/PPy/$SnO_2$ ternary composite film. The C and N, and O and Sn elements were the elements of the PPy and $SnO_2$, respectively, which were distributed uniformly throughout the whole area, indicating the homogeneous composites of PPy and $SnO_2$ were obtained. It should be noted that the Ag mapping image showed a wire-shape mapping, which was also confirmed by elemental analysis, revealed that the microwires were Ag.

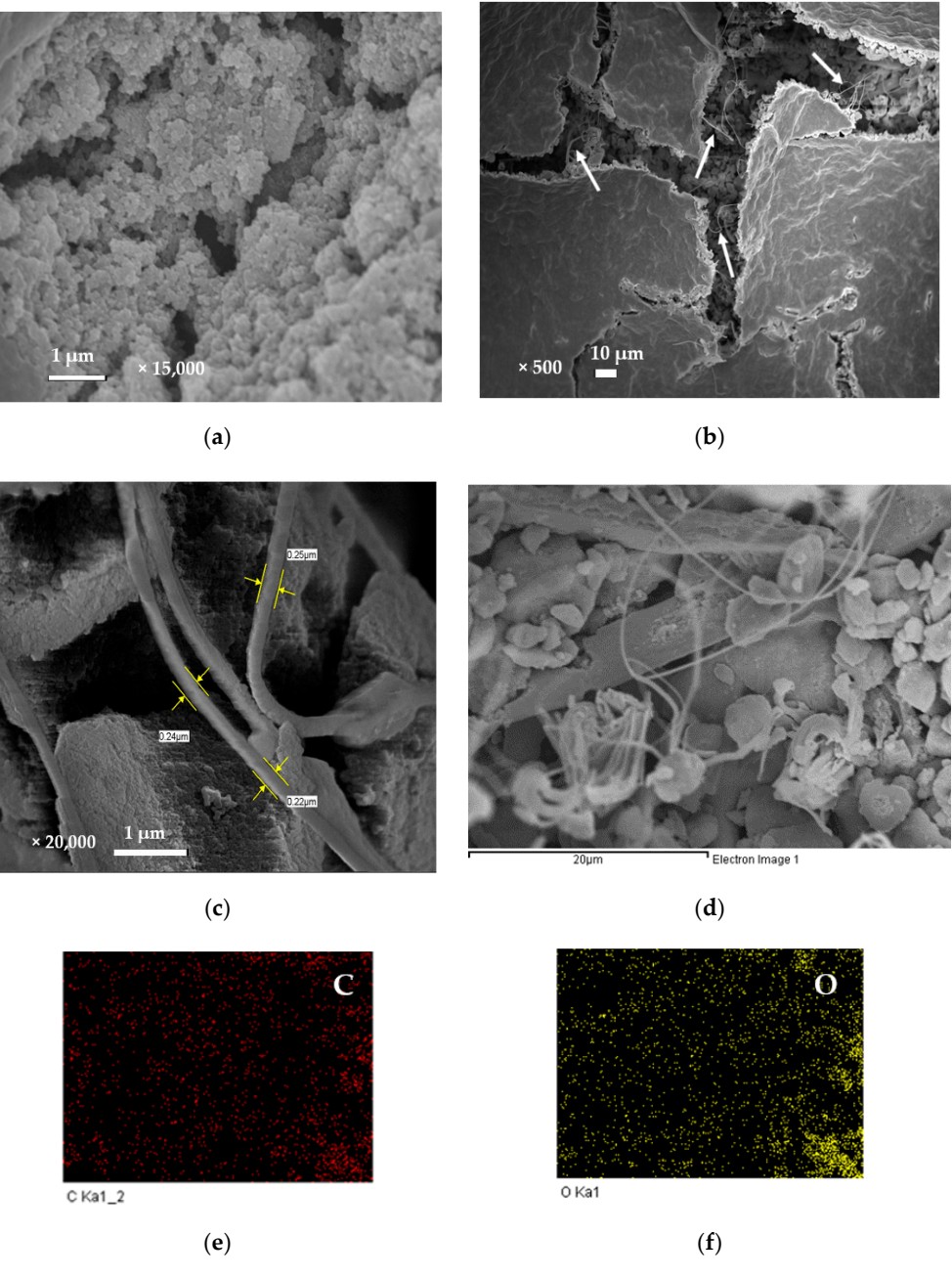

**Figure 4.** *Cont.*

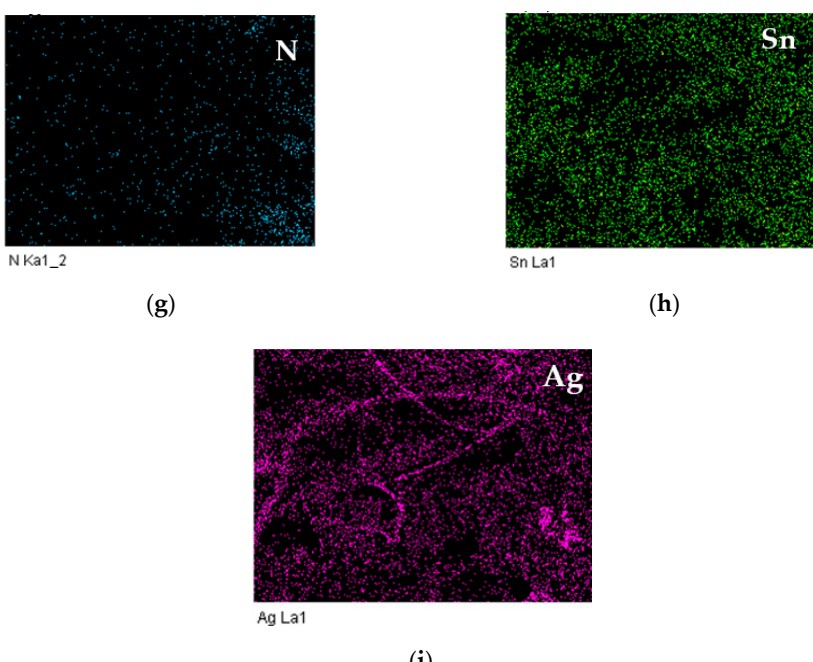

**Figure 4.** SEM images of (**a**) pristine $SnO_2$ film, (**b**) low-magnification image of Ag MWs/PPy/$SnO_2$ ternary composite film (white arrows indicated the Ag MWs), (**c**) high-magnification image of Ag MWs/PPy/$SnO_2$ ternary composite film, (**d**) image of the fracture in (**b**,**e**–**i**) its EDX mapping with elements for C, O, N, Sn, and Ag.

From the above discussions, and based on the combination of the reports of Li et al. [35], Lu et al. [36], and Balan et al. [38], a reaction mechanism of formation of Ag MWs/PPy/$SnO_2$ by one-step UV-induced photopolymerization can be proposed as follows: the novelty of the fabrication of a Ag MWs/PPy/$SnO_2$ ternary composite through a one-step photochemical process. The simultaneous photoreduction of silver and crosslinking photopolymerization of Py involved $SnO_2$ generated. The silver ions located at the silver/PPy/$SnO_2$ ternary composite and were reduced to silver atoms, then, favoring the nucleation and co-aggregation of silver particles that grew normally to generate silver microwires. At the same time, PPy was polymerized in situ on their surfaces. In this approach, photopolymerization and microcrystal formation occurred simultaneously. As the reaction continued, Ag MWs/PPy/$SnO_2$ ternary composite was obtained.

### 3.2. Electrical and Humidity-Sensing Properties of Humidity Sensors that were Made of Ag MWs/PPy/$SnO_2$ Ternary Composite Films

Figure 5 plots the effect of the addition amounts of loading of Ag and PPy on the impedance of Ag MWs/PPy/$SnO_2$ ternary composite films at various relative humidities. The measurements were made at 25 °C, 1 V AC voltage, and 1 kHz frequency. Table 1 presents results concerning sensitivity (slope) and linearity ($R^2$) of the fitting line for the humidity sensors. The sensitivity (slope) of the humidity sensors increased in the order pristine $SnO_2$ < Ag MWs (3 mg)/PPy (0.05 g)/$SnO_2$ < Ag MWs (12 mg)/PPy (0.2 g)/$SnO_2$ < Ag MWs (6 mg)/PPy (0.1 g)/$SnO_2$. The pristine $SnO_2$ had no obvious changes in impedance in the range of 10–20% RH because it exhibited high impedance (about 80 MΩ). Additionally, the pristine $SnO_2$ had only 1.5 order of impedance changes in the range of 30–90% RH. For reducing the high impedance and for improving the sensitivity and linearity of the pristine $SnO_2$ film for practical usage, a ternary composite material of Ag MWs/PPy/$SnO_2$ was prepared. Adding Ag MWs/PPy to pristine $SnO_2$ effectively reduced this high impedance and improved the sensitivity. The impedance decreased with increasing the amount of loading of Ag MWs/PPy. This result suggests that the new electrical conduction occurred by the Ag MWs/PPy loading. The impedance decreased as RH increased, and the range of humidity sensing increased with increasing the addition amounts

of loading of Ag MWs/PPy, enhancing the sensitivity of the Ag MWs/PPy/SnO$_2$ ternary composite. The sensitivity (slope) of the Ag MWs (6 mg)/PPy (0.1 g)/SnO$_2$ in the range 10–30%RH differed from that in the range 30–90%RH. The slope declined as the RH increased from 30 to 90%RH. This phenomenon was related to that the Ag MWs (6 mg)/PPy (0.1 g)/SnO$_2$ film (with added Ag MWs/PPy as electrical conduction) adsorbed the most water molecules, especially at low RH (10–30%RH), easily increasing the mobility of the solvated ions (H$_3$O$^+$), causing more sensitive in the 10–30%RH. At a higher RH (30–90%RH), fewer active sites are available for this reaction to occur, so the slope of the Ag MWs (6 mg)/PPy (0.1 g)/SnO$_2$ film declined. The sensitivity of the Ag MWs (12 mg)/PPy (0.2 g)/SnO$_2$ was lower than Ag MWs (6 mg)/PPy (0.1 g)/SnO$_2$. This phenomenon was related to that the Ag MWs (12 mg)/PPy (0.2 g)/SnO$_2$ exhibited low impedance below 30%RH, so that the impedance decreased slightly with an increase in %RH. The Ag MWs (6 mg)/PPy (0.1 g)/SnO$_2$ had the best sensitivity and acceptable linearity, so it was further tested to study its humidity-sensing properties and mechanism.

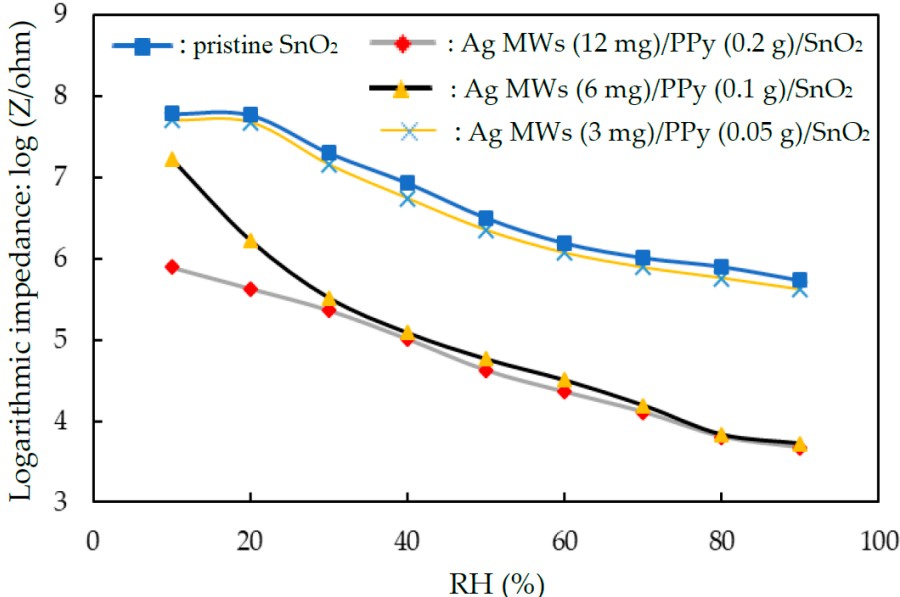

**Figure 5.** Impedance versus relative humidity for humidity sensors that were made of pristine SnO$_2$ and Ag MWs/PPy/SnO$_2$ in various compositions.

**Table 1.** Sensitivity and linearity of impedance-type humidity sensors that were made of pristine SnO$_2$ and Ag MWs/PPy/SnO$_2$ ternary composite films.

| Materials | Sensitivity [a] (log Z/% RH) | Linearity [b] ($R^2$) |
|---|---|---|
| Pristine SnO$_2$ [c] | −0.0259 | 0.9470 |
| Ag MWs (12 mg)/PPy (0.2 g)/SnO$_2$ | −0.0292 | 0.9932 |
| Ag MWs (6 mg)/PPy (0.1 g)/SnO$_2$ | −0.0407 | 0.9303 |
| Ag MWs (3 mg)/PPy (0.05 g)/SnO$_2$ | −0.0288 | 0.9581 |

[a] Sensitivity is defined as the slope of the of the linear fitting curve from 10 to 90% RH except SnO$_2$. [b] Linearity is defined as the R-squared value (correlation coefficient) of the linear fitting curve from 10 to 90% RH except SnO$_2$. [c] Linear fitting curve is from 30 to 90% RH.

Figure 6 plots the effect of the time of UV-irradiation photopolymerization on the sensitivity and linearity of the humidity sensor that based on Ag MWs (6 mg)/PPy (0.1 g)/SnO$_2$. The results concerning sensitivity (slope values) and linearity ($R^2$ value) of the fitted line at the tested RH values ranged from 10 to 90% RH were shown in Figure 6. No obvious variations in the sensitivity of the humidity sensor was observed when the time of the UV-irradiation photopolymerization increased from 20 to 60 min. This phenomenon was related to the fact that the UV was used to induce pyrrole polymerization in solution of AgNO$_3$ as oxidant, and the whole reaction of Ag MWs/PPy/SnO$_2$ ternary composite was

finished in 20 min [39], which was faster than the traditional wet chemical polymerization using $Fe^{3+}$ as an oxidant [51].

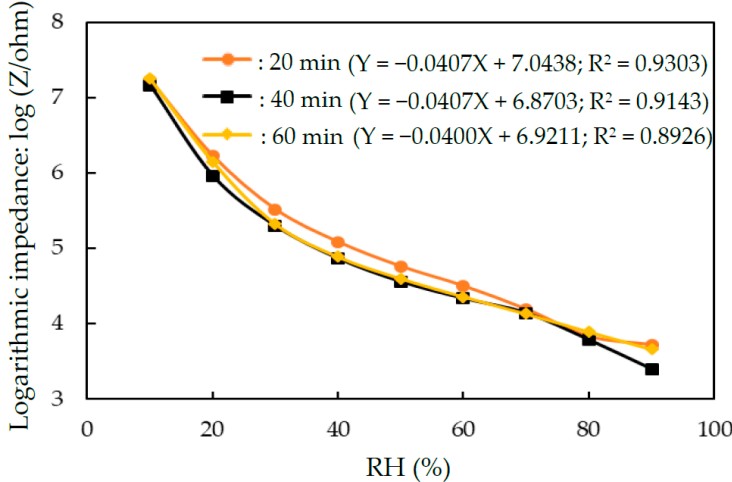

**Figure 6.** Impedance versus relative humidity for humidity sensors that were made of Ag MWs (6 mg)/PPy (0.1 g)/SnO$_2$ under various time of the UV-irradiation photopolymerization.

Figure 7a plots the hysteresis of the humidity sensor based on the Ag MWs (6 mg)/PPy (0.1 g)/SnO$_2$ ternary composite film. The average hysteresis over the RH ranged from 10 to 90% RH was below 1.8% RH in a dry-to-humid cycle. Figure 7b plots the effect of ambient temperature on the impedance of the humidity sensor based on the Ag MWs (6 mg)/PPy (0.1 g)/SnO$_2$ ternary composite film as function of RH. The impedance as function of the RH characteristic curve of the humidity sensor decreased with increasing the ambient temperature. This result was related to the fact the mobile charge carriers in the Ag MWs/PPy were induced by the heat energy. The average temperature coefficient was about −0.10% RH/°C over the humidity ranged from 10 to 90% RH. Figure 7c plots the response/recovery times of the humidity sensor based on the Ag MWs (6 mg)/PPy (0.1 g)/SnO$_2$ ternary composite film. The response/recovery times were 46/54 s. The long response/recovery times may be related to the large pore volume of the Ag MWs (6 mg)/PPy (0.1 g)/SnO$_2$ ternary composite film increased the stable adsorption of water molecules and resulted in increasing the response/recovery times. Thus, it should be efficient in decreasing the dimension of structures of the Ag MWs (6 mg)/PPy (0.1 g)/SnO$_2$ ternary composite film for enhancing the response/recovery kinetics in the future work. Figure 7d plots the effect of the applied frequency on the impedance of the humidity sensor based on the Ag MWs (6 mg)/PPy (0.1 g)/SnO$_2$ ternary composite film versus RH. It is obvious that the applied frequency affected more significantly the impedance of the humidity sensor based on the Ag MWs (6 mg)/PPy (0.1 g)/SnO$_2$ ternary composite film at low humidity (<40% RH) than that at high humidity. Figure 7e plots the long-term stability of the humidity sensor based on the Ag MWs (6 mg)/PPy (0.1 g)/SnO$_2$ ternary composite film. No obvious deviations in impedance at testing points of 10, 60, and 90% RH was found within 35 days. Table 2 compares the humidity-sensing properties of this work with those sensors based on SnO$_2$-based materials reported in previous literature [22–32]. The present humidity sensor based on the Ag MWs (6 mg)/PPy (0.1 g)/SnO$_2$ ternary composite film using one-step UV-irradiation photopolymerization technique exhibited a long humidity-working range, a comparable sensitivity and low hysteresis compared to humidity sensors that were made of ions (Li$^+$, K$^+$)-, Sb metal-, RGO- and metal oxides (ZnO, TiO$_2$, WO$_3$)-modified SnO$_2$ materials. These results were related to that the loading of embedded Ag MWs/PPy induced a new electrical conduction in Ag MWs (6 mg)/PPy (0.1 g)/SnO$_2$ ternary composite film for widening humidity-working range, causing more sensitivity.

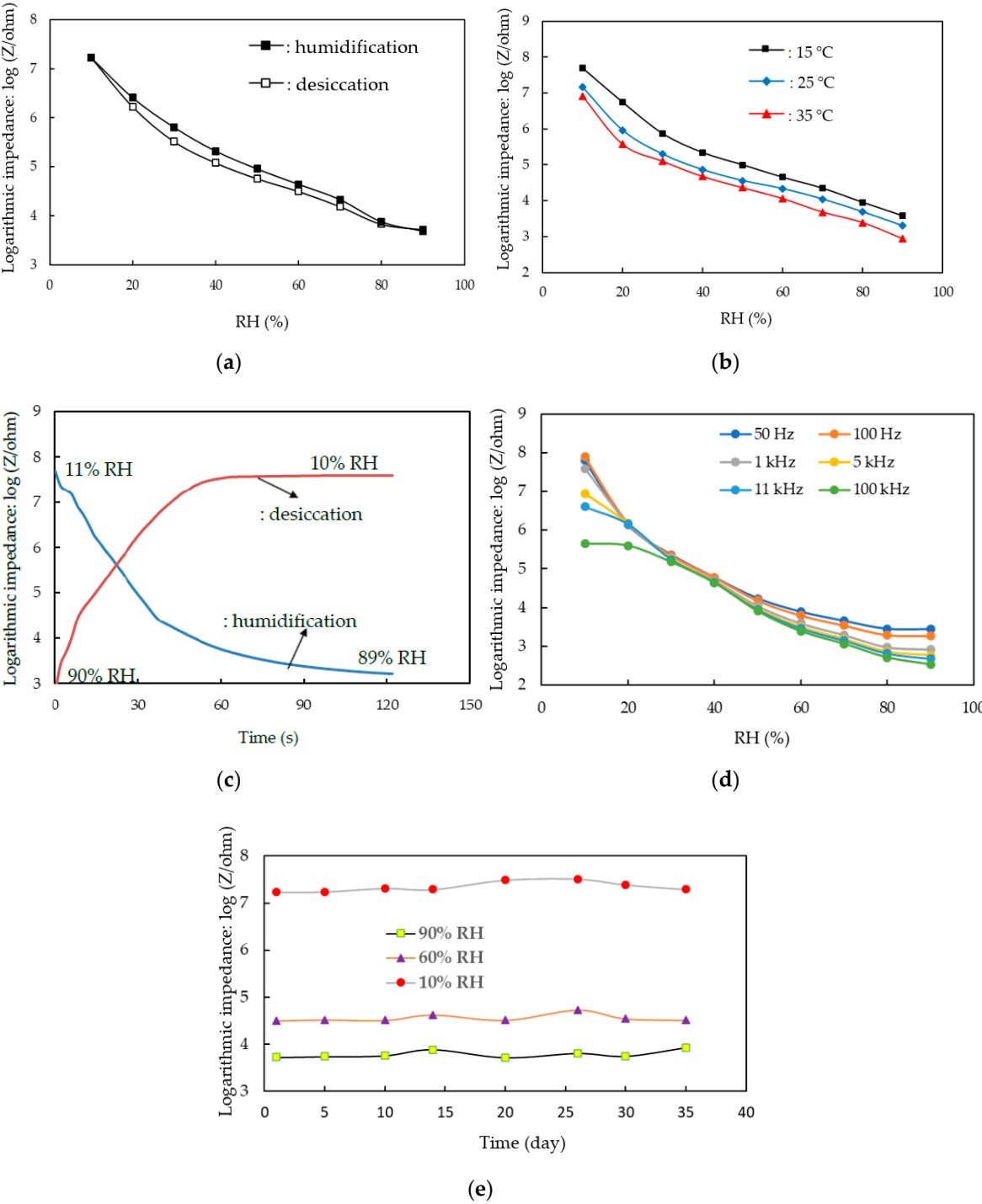

**Figure 7.** Humidity-sensing properties of the humidity sensor that was made of Ag MWs (6 mg)/PPy (0.1 g)/SnO$_2$ ternary composite film. (**a**) Hysteresis of the sensor, (**b**) effect of ambient temperature on the response of the sensor, (**c**) response/recovery times of the sensor, (**d**) effect of applied frequency on the response of the sensor, (**e**) long-term stability of the sensor.

**Table 2.** Humidity sensor performance of this work compared with the literatures based on the SnO$_2$-based humidity sensors.

| Sensing Material | Working Range (% RH) | Sensitivity | Hysteresis (% RH) | Response Time (s) | Ref. |
|---|---|---|---|---|---|
| Fe-doped SnO$_2$ | 11–95 | 6479.5 [a] | – | 1 | [22] |
| RGO/SnO$_2$ | 11–97 | 1604.89 pF/% RH | 1.8 | 102 | [23] |
| Ag-SnO$_2$/SBA-15 | 11–98 | – | 0.9 | 5 | [24] |
| ZnO/SnO$_2$ | 40–90 | 90.56 [b] | – | 3 min | [25] |
| TiO$_2$-SnO$_2$ | 20–90 | – | – | 20 | [26] |
| WO$_3$-SnO$_2$ | 10–90 | 2.63 MΩ/% RH | – | 121 | [27] |
| WS$_2$-SnO$_2$ | 0–95 | 23,900.4 [c] | – | 250 | [28] |
| Ag-doped SnO$_2$ | 11–98 | 4 order [d] | 1 | 4 | [29] |
| Sb-doped SnO$_2$ | 22–40 | – | – | – | [30] |
| KCl-doped SnO$_2$ | 11–95 | 4 order [d] | – | 5 | [31] |
| Li$^+$-doped SnO$_2$ | 33–85 | 36.3 [a] | – | 1 | [32] |
| Ag MWs/PPy/SnO$_2$ | 10–90 | 0.0407 [e] | <1.8 | 46 | This work |

[a] Sensitivity is defined as $I_{RH}/I_{dry}$. [b] Sensitivity is defined as $R_{40\% RH}/R_{90\% RH}$. [c] Sensitivity is defined as $R_{dry}/R_h$. [d] Sensitivity is defined as order in impedance changes over entire testing humidity range. [e] Sensitivity is defined as slope ($-(\log Z/\% RH)$) of the linear fitting curve over entire testing humidity range.

*3.3. Humidity-Sensing Mechanism*

Impedance spectroscopy was a powerful tool to analyze the conduction mechanisms of the humidity sensor based on the Ag MWs (6 mg)/PPy (0.1 g)/SnO$_2$ ternary composite film. The standard method to produce impedance spectra was to measure the impedance in frequency domain in the range of 50 Hz–100 kHz, RH in the range of 10–90% RH, at 1 V AC voltage and at 25 °C. Figure 8 plots the measured impedance spectra. The horizontal axis (Zr) and vertical axis (Zi) show the real component and the imaginary component of the impedance (Z), respectively. At a low humidity (20% RH), a similar semicircle plot of film impedance was observed (Figure 8a). This indicates that the film was modeled as an equivalent parallel circuit that included a resistor and a capacitor [52–54]. As RH increased (40%RH), a distorted semicircle and a line appeared in the low frequency range (Figure 8b) was observed, and the line became long when RH further increased to 80% RH (Figure 8c). These results indicate that the plot of impedance entered two regions: a semicircle (at high frequencies) was associated with the Ag MWs (6 mg)/PPy (0.1 g)/SnO$_2$ ternary composite film intrinsic impedance and an inclined line (at low frequencies) represented Warburg impedance, which was established by the diffusion of ions (protons ($H_3O^+$) across the interface between the electrode and the Ag MWs (6 mg)/PPy (0.1 g)/SnO$_2$ ternary composite film. From the obtained complex impedance spectra, a humidity-sensing mechanism of the present humidity sensor was proposed as follows: at low RH, the semicircular plot showed the intrinsic impedance of the Ag MWs (6 mg)/PPy (0.1 g)/SnO$_2$ ternary composite film; as increased RH, water molecules adsorbed on the Ag MWs (6 mg)/PPy (0.1 g)/SnO$_2$ ternary composite film, then gradually dissociated for forming $H_3O^+$ ions, in which these $H_3O^+$ ions migrated across the liquid-like Ag MWs (6 mg)/PPy (0.1 g)/SnO$_2$ ternary composite film to the interdigitated Au electrodes [53–55]. Therefore, the conduction process of this sensor can be attributed to $H_3O^+$ ions when water molecules were adsorbed.

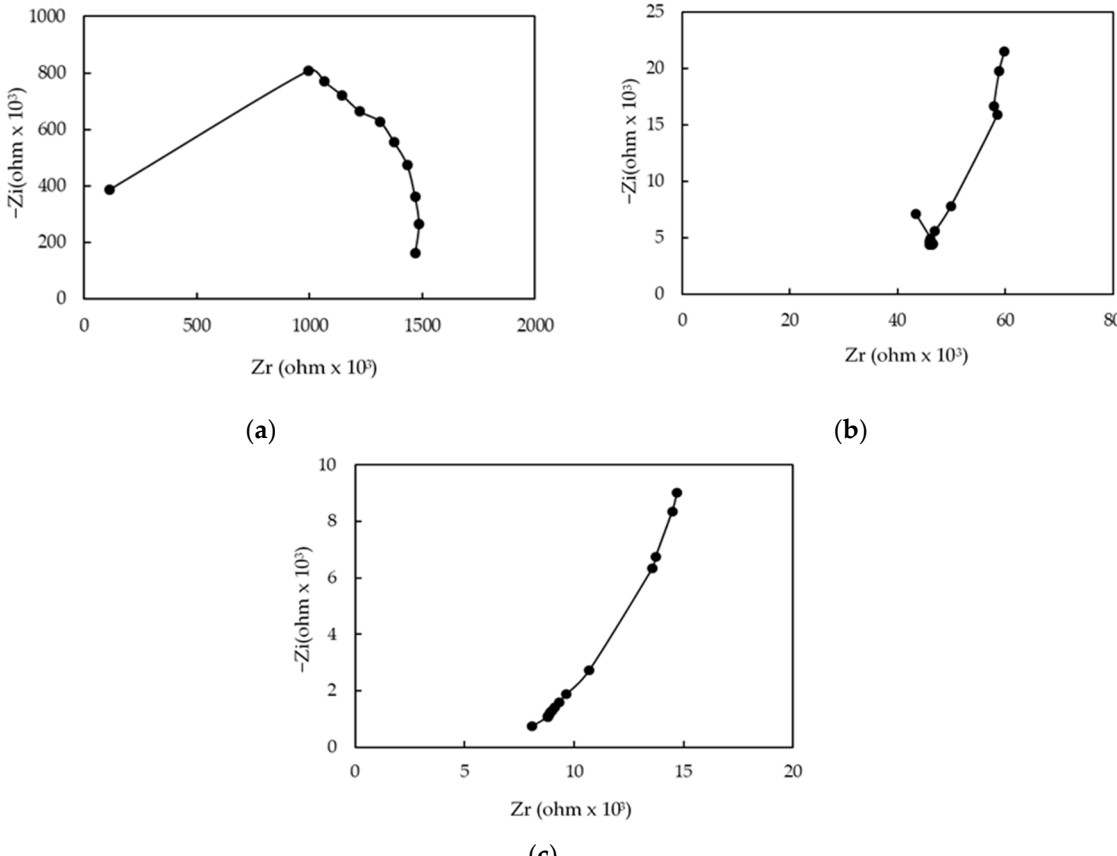

**Figure 8.** Complex impedance plots of humidity sensor that was made of Ag MWs (6 mg)/PPy (0.1 g)/SnO$_2$ ternary composite film at (**a**) 20% RH, (**b**) 40% RH, and (**c**) 80% RH.

## 4. Conclusions

The method of in-situ one step UV-induced photopolymerization, using AgNO$_3$ as an oxidant, on an alumina substrate was demonstrated to form an Ag MWs (6 mg)/PPy (0.1 g)/SnO$_2$ ternary composite film as a novel humidity sensor. The process of in-situ forming the Ag MWs in the Ag MWs (6 mg)/PPy (0.1 g)/SnO$_2$ ternary composite film was simple, cost-effective and timesaving. Moreover, the Ag MWs/PPy provided new conduction pathways in the Ag MWs (6 mg)/PPy (0.1 g)/SnO$_2$ ternary composite film, thus improving the conduction, sensitivity, and linearity of this sensing film. The humidity sensor based on Ag MWs (6 mg)/PPy (0.1 g)/SnO$_2$ ternary composite film had high sensitivity and acceptable linearity (Y = −0.0407X + 7.0438; R$^2$ = 0.9303), low hysteresis (<1.8% RH), low ambient temperature coefficient (−0.10% RH/°C), fast response/recovery times (46/54 s), and good long-term stability (at least 35 days). Based on the complex impedance spectra results, the H$_3$O$^+$ ions dominated the conductance of this humidity sensor.

**Author Contributions:** Conceptualization, P.-G.S.; Methodology, P.-G.S.; Investigation, P.-G.S., and P.-H.L.; Writing—original draft preparation, P.-G.S.; Writing—review and editing, P.-G.S.; Supervision, P.-G.S.; Project administration, P.-G.S.; Funding acquisition, P.-G.S. All authors have read and agreed to the published version of the manuscript.

**Funding:** This research was founded by Ministry of Science and Technology of Taiwan, grant no. MOST 109-2113-M-034-001.

**Conflicts of Interest:** The authors declare no conflict of interest.

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
