# Peer review of "Electrical and Humidity-Sensing Properties of Impedance-Type Humidity Sensors that Were Made of Ag Microwires/PPy/SnO2 Ternary Composites"

_chemosensors, doi:10.3390/chemosensors8040092_

Round 1
Reviewer 1 Report
This manuscript reports on humidity-sensing properties of impedance-type humidity sensors of Ag microwires/PPy/SnO2 ternary composites. Overall, the manuscript was well organized in terms of material synthesis, device characterization, and mechanism-related experiment. Even if experimental results proved the potential capability of composite sensor, the following issues should be addressed for the improved manuscript.
- The effect of loading Ag and PPy on the sensing property of the humidity sensing device was investigated by measuring the impedance of Ag/MWs/PPy/SnO2 ternary composite films. Specially, Ag MWs (6mg)/PPy (0.1 g)/SnO2 had the best sensitivity. In that case, however, different slope was observed for the range of 10~30% RH and 30~90% RH. Please add the reason why the phenomenon happened there.
- The numbers in the SEM image of Figure 3 should be modified for more clarity. The Figures should be also rearranged to one column or row in order to remove the empty space in the Figure set. The same rule should be applied to the Figure 4,7, and 8 which include a couple of empty space in each Figure set.
- In Table 2 for the comparison of sensing performance, sensitiviey and response time is not more better than other reference devices. Please add the reason for the poor performance and the methods to further improve the performance.
Author Response
Reply to reviewer 1’s comments:
- The effect of loading Ag and PPy on the sensing property of the humidity sensing device was investigated by measuring the impedance of Ag/MWs/PPy/SnO2ternary composite films. Specially, Ag MWs (6mg)/PPy (0.1 g)/SnO2 had the best sensitivity. In that case, however, different slope was observed for the range of 10~30% RH and 30~90% RH. Please add the reason why the phenomenon happened there.
Author reply:
Thanks the reviewer’s suggestion. The explanation about the results of the different slope of the Ag/MWs/PPy/SnO2 ternary composite film in the range of 10~30% RH and 30~90% RH was added in p. 6, lines 296 and 297 and p. 7, lines from 298 to 302.
- The numbers in the SEM image of Figure 3 should be modified for more clarity. The Figures should be also rearranged to one column or row in order to remove the empty space in the Figure set. The same rule should be applied to the Figure 4, 7, and 8 which include a couple of empty space in each Figure set.
Author reply:
Thanks the reviewer’s suggestion. Firstly, the SEM images is in Figure 4. For Figure 4, the images were rearranged to one column and row. Moreover, the numbers of the SEM images were modified for more clarity as shown in Figure 4 caption in p. 6, lines 267 and 268. The Figures 7 and 8 were also rearranged to one column and row.
- In Table 2 for the comparison of sensing performance, sensitivity and response time is not more better than other reference devices. Please add the reason for the poor performance and the methods to further improve the performance.
Author reply:
Thanks the reviewer’s suggestion. It is difficult to directly compare the sensitivity of various humidity sensors because the definition about the sensitivity in the literatures was different. Therefore, the clear definition about the sensitivity was shown as the footnote of the Table 2. Based on the impedance changes from low humidity to high humidity, we think the sensitivity of present humidity sensor in this work was comparable to the SnO2-based humidity sensors in the literature. Therefore, the “high sensitivity” was changed to “comparable sensitivity” in p. 8, line 389. The explanation about the poor performance and the methods to further improve the response/recovery times of the present humidity sensor was added in p. 8, lines from 374 to 379.
Reviewer 2 Report
This manuscript presents the humidity sensing properties of Ag MWs/PPy/SnO2 composites via impedance measurement. The approach provides promising properties for the potential implementation of the humidity sensors. However, the authors need to add a more straightforward explanation of specific results before the consideration of the acceptance.
- In the introduction section, the authors described the benefits of SnO2, PPy and Ag wires for a sensor device. However, it is unclear about the specific requirement to sense the humidity via impedance measurement. In other words, the manuscript may want to include quantitative information about impedance, surface area, a contribution from ions-dominating resistance change, etc. It appears that the novelty of the manuscript may lie in investigation of the sensing properties of a new combination of three materials. Besides, it is unclear about the reasons to choose SnO2 over TiO2 or other metal oxides for fabricating the composites or sensing the humidity.
- Figure 2: It would be better to present the phase of SnO2 in the XRD result.
- The white arrows in SEM indicate wires formed on sensing materials. Is it Ag MWs not Ag MWs and PPy composite? The authors may want to add a label in the figure caption.
- What are the reasons for the slope change in Figure 5? The authors explain the change of impedance with the addition of conducting Ag MWs and PPy. However, the mechanism to explain the slope change or the observed steep slope in Ag MWs (6 mg)/PPy (0.1 g) is unclearly explained.
- In page 8, the authors described “The present humidity sensor based on the Ag MWs (6 mg)/PPy (0.1 g)/SnO2 ternary composite film using one-step UV-irradiation photopolymerization technique exhibited a long humidity-working range, high sensitivity and low hysteresis compared to humidity sensors that were made of ions (Li+, K+)-, metals (Sb, Ag)-, RGO- and metal oxides (ZnO, TiO2, WO3)-modified SnO2” What are the supporting principles or reasons to present the aforementioned properties over other composites?
- In the Humidity-sensing mechanism section and Conclusion: The authors explained that the conduction process of this sensor can be attributed to H3O+ ions when water molecules were adsorbed. In the previous sentence, the authors mentioned different contributions to impedance (intrinsic at low RH and ionic with increasing RH). The discussion needs more clarification to explain the mechanism. Lots of articles had already described the change of conduction from electronic to ionic with RH levels. It is unclear whether the authors also want to claim the observation of similar behavior in their composite sensing materials.
Author Response
Reply to reviewer 2’s comments:
- In the introduction section, the authors described the benefits of SnO2, PPy and Ag wires for a sensor device. However, it is unclear about the specific requirement to sense the humidity via impedance measurement. In other words, the manuscript may want to include quantitative information about impedance, surface area, a contribution from ions-dominating resistance change, etc. It appears that the novelty of the manuscript may lie in investigation of the sensing properties of a new combination of three materials. Besides, it is unclear about the reasons to choose SnO2over TiO2 or other metal oxides for fabricating the composites or sensing the humidity.
Author reply:
Thanks the reviewer’s suggestion. The explanation about the reasons to choose SnO2 over TiO2 or other metal oxides for fabricating the composites or sensing the humidity was added in p. 1, lines from 36 to 40.
- Figure 2: It would be better to present the phase of SnO2in the XRD result.
Author reply:
Thanks the reviewer’s suggestion. The explanation about the phase of the SnO2 in the XRD was added in p. 3, lines 132 and 133.
- The white arrows in SEM indicate wires formed on sensing materials. Is it Ag MWs not Ag MWs and PPy composite? The authors may want to add a label in the figure caption.
Author reply:
Thanks the reviewer’s suggestion. Based the experimental results, it is Ag MWs. The explanation about the label for the Ag MWs was added in the Figure. 4(b) caption in p. 6, line 266.
- What are the reasons for the slope change in Figure 5? The authors explain the change of impedance with the addition of conducting Ag MWs and PPy. However, the mechanism to explain the slope change or the observed steep slope in Ag MWs (6 mg)/PPy (0.1 g) is unclearly explained.
Author reply:
Thanks the reviewer’s suggestion. The explanation about the results of the different slope of the Ag/MWs/PPy/SnO2 ternary composite film in the range of 10~30% RH and 30~90% RH was added in p. 6, lines 296 and 297 and p. 7, lines from 298 to 302.
- In page 8, the authors described “The present humidity sensor based on the Ag MWs (6 mg)/PPy (0.1 g)/SnO2ternary composite film using one-step UV-irradiation photopolymerization technique exhibited a long humidity-working range, high sensitivity and low hysteresis compared to humidity sensors that were made of ions (Li+, K+)-, metals (Sb, Ag)-, RGO- and metal oxides (ZnO, TiO2, WO3)-modified SnO2” What are the supporting principles or reasons to present the aforementioned properties over other composites?
Author reply:
Thanks the reviewer’s suggestion. These results were related to that the loading of embedded Ag MWs/PPy induced a new electrical conduction in Ag MWs (6 mg)/PPy (0.1 g)/SnO2 ternary composite film for widening humidity-working range, causing more sensitivity. The explanation was added in p. 8, lines from 391 to 393.
- In the Humidity-sensing mechanism section and Conclusion: The authors explained that the conduction process of this sensor can be attributed to H3O+ ions when water molecules were adsorbed. In the previous sentence, the authors mentioned different contributions to impedance (intrinsic at low RH and ionic with increasing RH). The discussion needs more clarification to explain the mechanism. Lots of articles had already described the change of conduction from electronic to ionic with RH levels. It is unclear whether the authors also want to claim the observation of similar behavior in their composite sensing materials.
Author reply:
Thanks the reviewer’s suggestion. The explanation about the humidity-sensing mechanism for Ag MWs (6 mg)/PPy (0.1 g)/SnO2 ternary composite film was rewritten for clearly reading and was added in p. 10, lines from 463 to 471.
Round 2
Reviewer 1 Report
The author tried to fully address the comments raised by the reviewers and then reflected the revision to the revised manuscript. The logic and structure are moderate as well. Thus, it can be published as it is.
Reviewer 2 Report
The authors carefully addressed the comments raised by this reviewer.